# Attitudes of at-risk older adults about prevention of cardiovascular disease and dementia using eHealth: a qualitative study in a European context

Ulrika Akenine [1], Mariagnese Barbera,[2] Cathrien RL Beishuizen,[3] Mandana Fallah Pour,[1,4] Juliette Guillemont [5] Anna Rosenberg,[2] Nicola Coley,[5,6] Francesca Mangialasche,[1,7] Lotta Salo,[2] Stephanie Savy,[5] A Jeannette Pols,[8] Sandrine Andrieu,[5,6] Edo Richard,[9,10] Hilkka Soininen,[2,11] Eric Moll van Charante,[3] Miia Kivipelto,[1,12,13,14,15] The HATICE study group

UA, MB and CRB contributed equally.

For numbered affiliations see end of article.

**Correspondence to**
Ulrika Akenine;
ulrika.akenine@ki.se

## ABSTRACT

**Objectives** Prevention of cardiovascular disease (CVD) and dementia is a key health priority among older adults. Understanding individuals' attitudes to, the prevention of these conditions, particularly when delivered through novel eHealth tools, could help in designing effective prevention programmes. The aim of the study was to explore the attitudes of older adults at increased risk of CVD and dementia regarding engagement in eHealth self-management prevention programmes, and to describe the facilitators and barriers.

**Design** A qualitative research approach was used. Data were collected through eight focus groups in Finland, France and the Netherlands. Data were analysed following the principles of grounded theory.

**Setting and participants** Forty-four community-dwellers aged 65+ at risk of CVD were recruited from a previous trial cohort in Finland, and through general practices in France and the Netherlands.

**Results** The study identified three categories: access to reliable information, trust in the healthcare providers and burden and stigma of dementia. A core category was also identified: the interactive process of the three categories influencing engagement in self-management prevention programme. The categories were interconnected through an interactive process and influenced by the local healthcare culture and context which shaped them differently, becoming either facilitators or barriers to engage in eHealth self-management prevention programmes.

**Conclusions** The study emphasises the importance of considering the interactions between the identified categories in this study, grounded in the local healthcare culture and context in further developments of eHealth self-management interventions that aim to prevent CVD and dementia.

**Trial registration number** ISRCTN48151589

### Strengths and limitations of this study

► This qualitative study benefits from an international setting focusing on older adults' attitudes regarding prevention of cardiovascular disease and dementia across the three West-European countries.

► The Healthy Aging Through Internet Counselling in the Elderly research team has expertise in qualitative, clinical and basic science research, ensuring an international perspective and a thorough understanding of the local healthcare settings.

► Due to the exploratory nature of the study and to ensure the richness of the data for data analysis, eight sessions of focus groups were conducted with 48 participants in total, in three West-European countries.

► Language barriers, as a limitation in the study, were decreased by aligning the research methodologies, applying an iterative process during data analysis and extensive discussions within the research group.

disease (CVD) and dementia has been reported,[1,2] causing health, economic and social burdens.[3] Prevention of CVD and dementia has been identified as a world-wide health priority.[2,4,5] Both CVD and dementia share several modifiable risk factors[1,2,6]—for example, hypertension, hypercholesterolaemia, diabetes mellitus, obesity, smoking, physical inactivity and unhealthy diet—providing the opportunity to test novel prevention interventions targeting both conditions.[7–9] Prevention of CVD and dementia among older adults is however complex requiring a combination of primary and secondary prevention, and even more challenging among those with existing comorbidity, or those labelled with

## INTRODUCTION

As the number of older adults increases worldwide, a rise in persons with cardiovascular

real diseases but no clear symptoms which makes the distinction between primary and secondary prevention unclear and difficult.[10] Promising novel prevention strategies, include eHealth, for its ease of access and of use.[11 12] However, engaging the general population with eHealth might be challenging,[13] and more insight is needed on how to maximise its advantages among older adults.

Successful prevention programmes are based on factors such as the selection of appropriate target populations, the implementation of optimal interventions, and using suitable delivery modalities.[14 15] Furthermore, engagement[16] and health literacy, and the ability to make sound decisions concerning health among the target population,[17] are crucial for the success of any public health intervention. Although risk factors have been identified, little is known about attitudes of older adults who are at risk of CVD and dementia about prevention initiatives, including eHealth. The design of prevention programmes tailored to this age group is therefore particularly challenging, and preventive trials among older adults are relatively scarce.[18] Such knowledge would be important for designing more effective preventive programmes and facilitating individuals' engagement.

The factors influencing engagement in prevention programmes have been partially investigated using qualitative approaches.[19–21] Previous studies suggested that a positive attitude of the participant is essential to implement effective preventive care,[19] which in turn promotes healthy cognitive ageing.[20] However, it is unclear how this positive attitude can be supported. A personal relationship between the healthcare provider and the patient seems to be beneficial,[21] but more evidence is needed on how to best encourage lifestyle self-management. Previous research from our research group has considered self-management and eHealth applications as promising strategies to support prevention.[22] In self-management, the individual takes the responsibility and lead to manage his/her risk factors, instead of the healthcare provider.[23 24] Applying eHealth has the potential to support self-management due to its advantages, such as suitability for health education, interactivity and monitoring.[25 26] Previous research found (1) establishing a relationship of trust, (2) managing awareness and expectations, and (3) appropriate timing and monitoring of the process of behaviour change, as important to support an effective behaviour change in prevention of CVD and cognitive decline.[22] In addition, the previous literature has not simultaneously considered both CVD and dementia, and international studies are still scarce. Factors related to the country-specific context might considerably impact individuals' perception of prevention.[27] For example, accessing novel tools for healthcare can be perceived as a challenge, especially in areas where new technologies are not well established.[11]

The present study is part of the Healthy Aging Through Internet Counselling in the Elderly (HATICE) project.[28] The HATICE project tested the efficacy of an eHealth multidomain intervention, including a coach-supported internet platform[29] to improve older adults' self-management of risk factors for CVD and dementia, in a European randomised controlled trial (RCT). The RCT was carried out in Finland, France and the Netherlands. The aim of this substudy was to explore the attitudes of older adults at increased risk of CVD and dementia regarding engagement in eHealth self-management prevention programmes, and to describe the facilitators and barriers.

## METHODS

### Design

The study applied a qualitative research approach following the principles of grounded theory,[30] and was structured in sequential steps of data collection performed in three rounds of focus groups and analyses (figure 1). The study benefited from an international research group in the HATICE project[31] providing the knowledge and expertise in qualitative, clinical and basic science research, and ensuring an international perspective with an in-depth understanding of the local healthcare settings.

The study was conducted following the Consolidated criteria for Reporting Qualitative research.[32]

### Participants

Forty-four older adults at risk of CVD were purposively recruited from (1) a previous trial cohort[7] (Finland) and (2) through general practices (France, the Netherlands) (table 1). In order to recruit a population as similar as possible to the one that would have been recruited in the actual trial, participants were recruited using a simplified but comprehensive version of the inclusion criteria applied in the HATICE RCT.[28] The criteria were defined as follows: (1) age≥65 years, (2) basic internet literacy defined as use of email, (3) ≥2 self-reported cardiovascular risk factors defined as: hypertension (diagnosis or medication prescription), dyslipidaemia (diagnosis or medication prescription), active smoking and lack of physical exercise defined based on the WHO guidelines (at least 150 min of moderate intensity exercise per week) and/or (4) self-reported history of CVD (stroke/transient ischaemic attack, myocardial infarction, angina pectoris and/or peripheral arterial disease), (5) self-reported diagnosis of diabetes mellitus.

### Setting

Due to the international setting of the present study, different primary healthcare systems are relevant to be described shortly. In Finland, primary healthcare is provided in healthcare centres to which people are automatically assigned based on the address of residence. Their size, both in terms of patients cared for and catchment area, can vary significantly due to large differences in population density,[33] which can lead to difference in the service provided for non-emergent cases and prevention programmes. Nurses are the 'gatekeeper' of the system

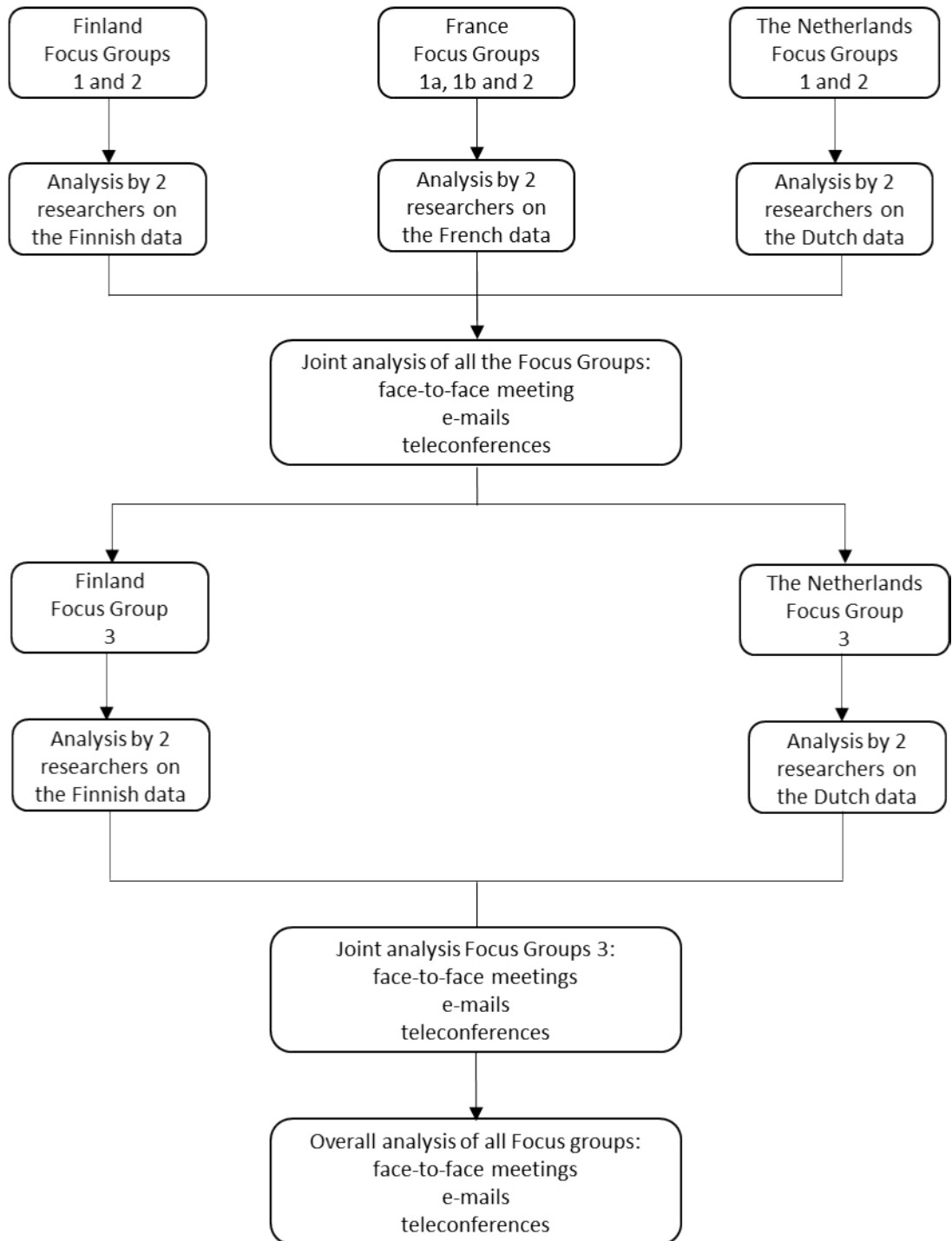

**Figure 1** Different stages of data collection and analysis.

and one of the main figures in relation to prevention. The patients are not assigned to a specific general practitioner (GP).[34] In France, GPs usually work in 'solo' practice and citizens can choose to which practice to register based on availability.[35] Patients are, therefore, followed by the same physician and same-day appointments are usually available for non-emergent cases. Nurses can provide care, mostly preventive, tertiary or palliative care, but only on GP prescription. They are also involved in screening programmes and health education.[36] As in Finland,

primary healthcare in the Netherlands is provided within healthcare centres that are, however, more evenly distributed in a country with a much higher population density compared with Finland. Patients register with a specific GP, making the doctor–patient relationship consistent and, to some extent, similar to the French system. However, nurses in the Netherlands have somehow a more independent role (eg, carry out specific consultations, such as for diabetes care and cardiovascular risk management and can prescribe certain types of medications).[37]

**Table 1** Summary demographics of the participants

| Country | Finland | | | France | | The Netherlands | | |
|---|---|---|---|---|---|---|---|---|
| Focus group session | 1 | 2 | 3 | 1 | 2 | 1 | 2 | 3 |
| N | 9 | 9 | 10 | 9 | 7* | 10 | 5* | 6 |
| Average age, mean | 70.4 | | 66 | 72.3 | 71.1 | 69.0 | 69.2 | 83.7 |
| Gender, n (female/male) | 3/6 | | 8/2 | 3/6 | 2/5 | 4/6 | 2/3 | 3/3 |
| Participants with at least a secondary education degree, n | 4 | | 3 | 8 | 6 | 6 | 4 | 6 |
| Participants with a history of CVD, n | 3 | | 2 | 4 | 3 | 2 | 2 | 2 |
| Internet literate participants | 9 | | na | 9 | 7 | 10 | 5 | 6 |

*The participants of these focus groups are a subset of the participants who attended the previous session.
CVD, Cardiovascular disease; na, not asked.

## Data collection

Eight focus group sessions structured in three rounds (figure 1) were conducted between October 2013 and September 2015. A semistructured focus group interview guide was used,[38] including three sessions in Finland, three in the Netherlands and two in France. The interview guide comprised the main topics and subtopics focusing on participants' attitudes regarding engagement in eHealth self-management prevention programmes, and the facilitators and barriers (box 1). Two members of each local research team were present at each session (average duration of 2 hours): (1) an experienced moderator in qualitative research and (2) an assistant to take notes. The meetings were tape recorded and transcribed verbatim, except for the first meeting in Finland and the second in France, due to technical issues in the tape recording. The researchers' detailed notes from these two sessions were therefore analysed instead.

## Data analysis

The data were analysed following the principles of grounded theory.[30] In each country, two researchers independently identified codes, combined and compared them in the axial-coding phase, created the categories, which were confirmed through consensus. The analyses were, therefore, performed in the local language, except in Finland, where neither of the researchers were native speakers. The transcripts and notes were first translated into English and independently cross-checked by two team members, both Finnish native-speakers and English fluent-speakers. After completing the analyses of the first six sessions in each country, the findings were combined in English, and further discussed and compared by the full team in a face-to-face meeting. To ensure that the results were grounded in the data and focused on interactions between the categories, the analyses formed an iterative process, transitioning from the original data to the categories. Moreover, a third round of focus groups was organised in the Netherlands and in Finland to enrich the data and get a better understanding about 'prevention in general', 'prevention of CVD', 'prevention of dementia' and 'individuals'' perspectives on 'eHealth self-management prevention programmes'. To increase the credibility of the findings, the analysis was extensively discussed within the research group, both at a national and a summary of the conclusions of the third focus group was returned to the participants to check for trustworthiness of the data. International level until agreement was reached.

## Patient and public involvement

Patients were not involved in this study. However, the participants were older adults at risk of CVD and dementia who were directly involved in the development of the HATICE eHealth application by taking part in focus groups in this substudy (see inclusion criteria in the Methods section).

## RESULTS

Three categories were identified, and a core category was developed, representing the the attitudes of older adults at increased risk of CVD and dementia regarding engagement in eHealth self-management prevention programmes, and the facilitators and barriers (see figure 2).

### Access to reliable information about CVD and dementia

Participants experienced confusion regarding the general meaning of prevention, and in particular prevention of CVD and dementia. Prevention was generally described as acting to avoid a certain disease, but when discussing how prevention should be put into practice, participants focused mostly on how to identify symptoms and when to initiate a treatment. Unawareness of the right time to act was described as the main barrier to engage in prevention. Participants could name risk factors for CVD and dementia, but they could not concretely explain how to manage them.

> It is important to recognise the symptoms [...]. For example, chest pain. [...] Make sure you visit the doctor in time and know what you should be alerted of. (fg1, Dutch participant)
>
> (fg = focus group)

**Box 1    Summary of focus group interview guide including main topics and subtopics**

Focus group round 1
**Prevention of CVD and dementia**
Knowledge about prevention.
Role of prevention in the development of CVD and dementia.
**Internet use**
Participants' use and attitude towards the internet.
Requirements for a user-friendly website.
Preventive programmes delivered through the internet.
**Relationship with the healthcare provider**
Important factors to receive good guidance by the healthcare provider.
Important factors to establish a good relationship with the healthcare provider.
**Focus group round 2**
**Layout of the HATICE platform***
Suggestion to make the layout more user friendly.
Feedback on the monitoring functionalities.
**Role of the coach in the HATICE intervention**
Preferred mode of communication and frequency of contacts.
Right for the coach to access the participants' information on the platform.
Role of the coach in the goal setting process.
**Peer support in the HATICE platform**
Attitude towards a peer internet forum.
**Self-support in the HATICE intervention**
Information about CVD and dementia required on the platform.
What type of news items/newsletter the participants will be interested in.
Digital rewards automatic reminders and comparison with peers as motivators.
**Focus group round 3**
**Prevention**
What prevention is, when it should start, who benefits, importance of genetic predisposition.
Can CVD be prevented? How? Risk factors, barriers and motivators.
Can dementia be prevented? How? Risk factors, barriers and motivators.
Comparison between CVD and dementia and their prevention.
**Motivation for prevention**
Motivating factors to act on for prevention.
Barriers to act on for prevention.
How the differences between CVD and dementia affect motivation for prevention?
How the differences between CVD and dementia affect the relationship with the healthcare provider?
Different attitudes of individuals and society to dementia, compared with CVD.
The role of fear.

CVD, Cardiovascular disease; HATICE, Healthy Aging Through Internet Counselling in the Elderly.
*The platform had not been yet finalised at the time of the second round of focus groups, therefore a preliminary version was shown

In all three countries, participants expressed a need for reliable information about CVD and dementia, and on how to put general recommendations into practice. Access to reliable information was regarded as essential to empower them towards prevention, and their lack of knowledge about these diagnoses represented a significant barrier to take action. They described reliable

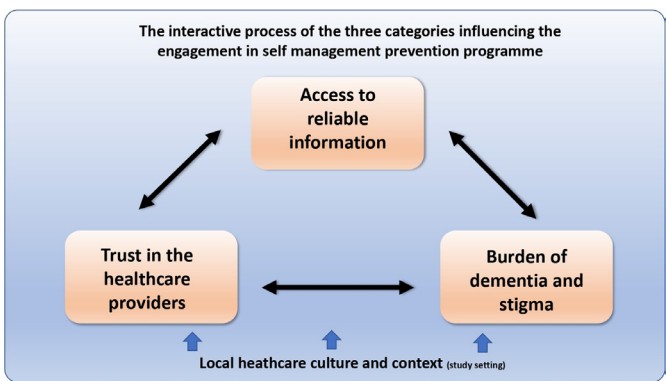

**Figure 2** Presentation of the core category and interactions between categories.

information as comprehensive information that is easy to understand, tailored to each individual situation and provided by a trustworthy source.

> [We need] a reference [...], a website with a search engine, information on diseases, up to date, specific. (fg1, French participant)

Not being able to distinguish trustworthy from untrustworthy sources was identified as another barrier. This was especially true for health-related information received from the Internet. The participants stressed that an eHealth platform, including support from a caregiver, can provide trustworthy information by authorised sources, tailored to individual needs.

> Young people surf and search much more on the Internet than the elderly. The elderly have much more difficulty to judge what information is of good quality and what isn't. (fg1, Dutch participant)

### Trust in the healthcare provider

The participants described that trust in their primary healthcare providers, including their GPs and nurses, as well as trust in the received health-related information and the healthcare system, is a prerequisite to engage in prevention programmes. Some participants mentioned that in order to provide the best possible medical advice, healthcare providers need to know their patients personally.

The participants referred to trust in different ways. Participants in France and the Netherlands highlighted the importance of having a good relationship with their primary healthcare providers, being one of the most important motivators to engage in prevention programmes including those delivered through eHealth tools. In fact, when discussing the HATICE platform, the Dutch and French participants expressed concerns regarding the interference with the regular healthcare provider who, to ensure continuity of care, expressed a strong preference for an eHealth prevention programme managed by their own primary healthcare provider.

You would prefer to go to your own GP who knows you already since so many years, rather than commit to someone you do not know. (fg3, Dutch participant)

Although the Finnish participants did not stress this aspect, they acknowledged the role of trust in the eHealth lifestyle coach and in data integrity when managing personal information as encouraging factors to actively participate in such eHealth prevention programmes.

Trust and expertise [are the most important qualities] (fg1, Finnish participant)

While Dutch and French participants strongly relied on their GPs and took their advice seriously, Finnish participants described themselves as more independent and critical towards medical advice. The Finnish participants stressed the importance of their autonomy and own responsibility for their health and prevention, expressing their pronounced interest in health self-management.

If your doctor prescribes medication you take it. After all, he has your best interest at heart. Even if you don't want to take them. (fg3, Dutch participant)

[Prevention is the] patient's responsibility for him/ herself. (fg1, Finnish participant)

### Burden and stigma of dementia

The participants compared the possibilities for prevention of CVD with those of dementia. They described CVD as having good treatment options and a possibility to recover, compared with dementia as a condition with no possibility for recovery. Participants associated feelings of fear, shame and hopelessness in anticipation of developing dementia.

You cannot reverse dementia, and from a cerebral infarction or heart infarction you can still recover. (fg3, Dutch participant)

The participants described that these feelings, in addition to the lack of an effective treatment, renders dementia a great burden, perceived as almost a 'death sentence'. They described that the burden was caused by the loss of one's independence, due to deterioration of cognitive skills, loss of physical capabilities, and loss of social relationships. Dementia was also regarded as a burden for the families, relatives and social relations of persons with dementia and society.

You see people with Alzheimer's disease […], you say let's hope my life won't end like that, like a vegetable. (fg1, French participant)

[Dementia requires [providing care by relatives and institutional care. It's an expensive disease […]. We do have a heavy burden to bear. (fg3, Finnish participant)

Participants generally expressed a pessimistic attitude towards prevention of dementia as opposed to CVD.

However, being physically, mentally and socially active was described as a potential preventive factor.

If you are using your brain, then, it has to do with, uh, postponing it. (fg3, Dutch participant)

Fear was described by the participants as an encouraging factor to engage more in the prevention of dementia than of CVD.

The fact that dementia shares many risk factors with CVD was not generally known or expressed by the participants. Concerning dementia, the Finnish participants' main concern was identifying the first symptoms and the right time to seek medical advice.

There should be information on when I should go to a doctor for tests. People always say that you should go on time, but I don't know when is on time. (fg1, Finnish participant)

Nonetheless, the key role of genetics that they attributed to dementia was closely linked to their scepticism towards its prevention.

It's a matter of wait and see. You can try to take preventive measures, but you cannot stop it (Dementia). If you are born to get dementia, you will get it eventually. You can take medications etc., then you can maybe delay the onset of the disease, but eventually it will catch you. (fg1, Dutch participant)

Dementia seems to be associated with stigma. This stigma represented a barrier to obtaining reliable information for the Finnish participants, who stated that the fear of dementia made it more difficult to talk about it compared with CVD, and to consult a doctor about it.

The fear can make you freeze and cause you to not be able to talk. Just wondering about it in your mind, not putting it into words. (fg3, Finnish participant)

### The interactive process of three identified categories influencing engagement in the self-management prevention programme

From the three categories, a core category was developed: the interactive process of the three identified categories influencing engagement in the self-management prevention programme. The three categories were interconnected through an interactive process and were strongly influenced by the local healthcare culture and context which shaped them differently (see figure 2). This interactive process is presented in figure 2. Figure 2 shows that in order to minimise the stigma, there is a need to receive relevant, reliable information, and to trust in healthcare providers. However, the burden and stigma of dementia was described as a barrier to receiving reliable information and trusting healthcare providers. The participants described that fear of dementia made it difficult to talk about it (eg, to GP), which in turn reinforced their perception of insufficient reliable information about the disease, which made it more frightening and unsafe. It

was also described that this fear was experienced more strongly for dementia compared with CVD. On the other hand, participants mentioned that the fear of dementia can improve their motivation to engage in dementia prevention programmes.

> I realised when another person got sick, I woke up then 10 years ago when my mother was diagnosed with Alzheimer and I panicked and realised that I must do something. (fg3, Finnish participant)

The analyses also demonstrated that the category of burden and stigma of dementia was also interconnected with the category of trust in healthcare providers; only if participants could trust the source of information, then they were likely to take action for their engagement in a prevention programme. The participants in all three countries described how the local healthcare system and context influenced their experiences in obtaining reliable health information, trust, and also in relation to the burden and stigma of dementia. For example, Finnish participants experienced a sense of responsibility for their health and disease prevention. They described their interest in health self-management related to specific diseases. Whereas, French and Dutch participants focused on the overall health status. The Finnish participants mostly highlighted the importance of trusting the source of information and the possibility of obtaining information independently. Dutch and French participants emphasised the importance of a trustworthy relationship with healthcare providers (eg, GPs and nurses), which led them to trust the information received. Personal responsibility for their health (eg, receiving reliable information about prevention of CVD and dementia) was important for the Finnish participants. Finnish participants described that they were habituated to independently make medical decisions, and critically question medical recommendations. In contrast, the French and Dutch participants relied more on the advice provided by their healthcare providers without questioning them. The importance and benefits of dementia prevention were more clearly acknowledged by the Finnish participants; however, Dutch participants believed more in luck and chance regarding disease prevention.

> We bicycled a lot and we liked / enjoyed doing that. But nevertheless, you can get the most horrible things. At the end of the day you have no influence on it. (fg3, Dutch participant)

## DISCUSSION

In this European qualitative study, the aim was to explore the attitudes of older adults at increased risk of CVD and dementia regarding engagement in eHealth self-management prevention programmes, and to describe the facilitators and barriers. Three categories were identified from the analysis: (1) access to reliable information, (2) trust in healthcare providers and (3) burden and stigma

of dementia. From these categories a core category was developed: The interactive process of the three identified categories influencing engagement in self-management prevention programmes. The three categories were interacting with each other and were influenced by the local healthcare culture and context, becoming either facilitators or barriers to engage in eHealth self-management prevention programmes. This interactive process suggests that in order for individuals to actively engage in the eHealth prevention programmes, there is a need for having access to reliable information about prevention of CVD and dementia. Providing reliable information regarding the prevention of these two conditions, from trustworthy sources, is therefore regarded as an opportunity for eHealth programmes to fulfil this need. However, in order to translate this information into knowledge and for it to be used by individuals who can take action towards engaging in prevention programmes, individuals need to trust the information provided and have a relationship of trust with their healthcare provider. The eHealth prevention programmes might therefore include support from individuals' own healthcare providers, or from online health coaches. The interactive process between categories stresses that trust in the healthcare provider, access to reliable information and the generated knowledge could support individuals and decrease their experience of stigma and burden. This interactive process can encourage individuals to take action towards engaging in prevention programmes (see figure 2). Previous research focusing on prevention of CVD from the perspective of nurses[22 39] and of older adults[21 40] identified the importance of establishing a relationship of trust, managing awareness and expectations (including individuals' level of knowledge) and providing personally tailored support. Previous research from our research group has also stressed the importance of considering the local healthcare practices to plan new forms of preventive healthcare involving the individual's self-management.[22]

Access to reliable information was identified as a key prerequisite for individuals' engagement in prevention programmes. The internet, however, was perceived as a confusing source of health-related information. The findings stress the importance of implementing prevention programmes, such as eHealth self-management programmes (eg, HATICE), administered by trustworthy organisations to provide reliable information about prevention. Previous research found that information from the Internet and media are a facilitator of stimulating engagement in preventive care from the perspective of healthcare providers (eg, GPs).[19]

The findings stress that trust in healthcare providers, the source of information, and the healthcare system are crucial facilitating factors. The lack of trust might hinder individuals from contacting their healthcare provider, following recommendations, receiving information and meaningfully translating the information into practice. The participants expressed that they have limited knowledge regarding concrete methods to prevent CVD and

dementia. Although dementia has been identified as a key public health priority worldwide,[41] knowledge about it, and its risk factors is limited in the general population.[42] Based on our results, the lack of knowledge about dementia among individuals might lead to fear, which hinders individuals from seeking help from the healthcare system and accessing the available opportunities for prevention, as well as a stigma, which might mirror the general attitude of the society.[43] Interestingly, the burden and stigma of dementia was identified both as a facilitator and a barrier for engagement in prevention. It can be assumed as a factor that might hinder individuals from contacting healthcare providers and receiving thorough information regarding the disease, including the current evidence on beneficial preventive interventions. It can also motivate them to engage in prevention and to take action. According to the literature, the concept of stigma is defined as 'having some form of mark or sign that denotes disgrace or discredit',[43] which refers to 'marked differences from what is 'normal' for a group of people, and to negative emotional and/or behavioural responses to those differences'.[44] When discussing prevention, prognosis and treatment of dementia, in contrast to CVD, the participants had a rather 'white-and-black' perspective, which is in line with the previous literature,[45] describing a belief that 'nothing can be done', which impacts individuals' well-being.[44] However, despite their pessimistic attitude regarding dementia prevention, the participants described how their fear of the disease was experienced as a factor encouraging them to actively engage in prevention programmes for dementia, compared with CVD, by being physically, mentally and socially active. Our findings regarding the dynamic interactions between the identified categories is supported by the previous literature, which highlighted the importance of education and information in reducing the fear and stigma associated with dementia.[45] Furthermore, the identified categories confirm previous findings on the urgent need, across all levels of society, for increased awareness and understanding of dementia (diagnosis, symptoms, treatment, risk factors and prevention), to improve the quality of life among older adults.[41 42]

The results highlight not only the similar attitudes about prevention of dementia and CVD across the three countries, which emerged as the three categories on a general level, but also the variations that are grounded in the cultural and contextual backgrounds that shaped the local healthcare culture and context in these three countries. Culture is defined as the beliefs, perceptions, values, norms, customs and behaviours that are shared by a group or society and are passed from one generation to the next through both formal and informal education.[46] In this study, context is intended as 'social environment', that is the pool of structures and social systems through which society is organised (eg, the healthcare system).[47] Engaging participants is key to successful prevention programmes. Although previous studies[19-21] investigated the motivating factors to engage older adults in

prevention programmes, none of these studies had an international design. This study emphasises how the local healthcare culture and context might affect individuals' needs and access to information, their trust in healthcare providers and their perceived dementia-related stigma and burden, which represents their attitudes regarding prevention. Previous research highlights how culture and context can influence manifestations of dementia.[48] It also stresses the importance of culture and context in shaping several aspects of caregiving, as well as public policies, to improve the awareness and understanding of dementia.[49] In this respect, one important difference identified in the study among the participants was that the Finnish participants were more independent and described their self-responsibility when making health-related decisions, whereas the Dutch and French participants relied more on their healthcare providers. These differences were also identified in a recent qualitative study from our research group conducted with Finnish and Dutch nurses about optimally supporting patients in CVD preventive care.[22] The study indicated that even if aims in preventive care were very similar between Finland and the Netherlands, patient empowerment and autonomy received more attention in Finland than the Netherlands.[22] This is a difference to the present study which may be due to differences in the three healthcare systems. Differences may be based on a more stable doctor-patient relationship in France and the Netherlands, as opposed to Finland, where a patient is not registered with a specific GP. Furthermore, Finland has a long history of intensive preventive programmes targeting CVD,[50] which made the concept of prevention well assimilated within the society. Finally, the Finnish participants were recruited within a previous preventive trial cohort.[7] Beishuizen et al[22] demonstrated that when designing and introducing new preventive healthcare applying eHealth self-management, local healthcare practices are to be considered to fulfil optimal engagement. According to the literature, these differences can be referred to as differences in organisation, their focus, accessibility, role of primary care and patient autonomy,[22] and national guidelines for primary and secondary prevention care.[10] Healthcare-related differences among the three countries were considered in the HATICE intervention design,[10] and these results confirm that context diversity should be considered when planning international prevention programmes. Further studies are required to identify the most effective preventive strategies across cultures.

### Methodological considerations
Although the study was conducted in small sections of each country's populations, these are representative samples of their respective areas, ensuring a fully international setting. Data collection in three different languages was challenging, as the translations into English might have influenced the results, and some of the nuances could have been lost in translation. However, interview guides were prepared with great care, including

consideration of language-related issues. The findings were extensively discussed within the teams, both at a national and international level. Moreover, the total number of focus groups conducted and the broad areas of expertise of the research team were a significant asset. Additionally, frequent feedback from all team members was instrumental for the good execution of the study and for the mutual understanding of the local settings. Differences in recruitment, such as the older age of the Dutch participants in the third round of focus groups and the enrolment of the Finnish participants from a previous preventive trial cohort[7] might have also affected our findings. In order to check the trustworthiness of the data, a summary of the conclusions of the third focus group was sent to the participants. The main reason for performing the member check in this way was that the third round of focus groups was covering all the areas of focus in the study and was complementary to the data from the first two rounds.

## CONCLUSIONS

The study identified three categories and a core category. The categories were interconnected through an interactive process and influenced by the local healthcare culture and context which shaped them differently, either as facilitators or barriers to engage in eHealth self-management prevention programmes. The findings can be integrated into future developments of eHealth self-management interventions to prevent modifiable risk factors for CVD and dementia. eHealth self-management programmes can fulfil the need for reliable and trustworthy health information. If a safe and trustworthy online environment can be developed, this may enhance engagement in prevention programmes and stimulate destigmatisation of dementia. The findings highlight the importance of taking the local healthcare culture and context into account when planning international prevention programmes. Studies on the perception of prevention and lifestyle changes during and after clinical trials among individuals at risk for, or at an early stage of, cognitive impairment, can provide further insights.

**Author affiliations**
[1]Division of Clinical Geriatrics, Center for Alzheimer Research, Department of Neurobiology, Care Sciences and Society, Karolinska Institutet, Stockholm, Sweden
[2]Institute of Clinical Medicine, Department of Neurology, University of Eastern Finland, Kuopio, Finland
[3]Department of General Practice, Amsterdam UMC/University of Amsterdam, Amsterdam, The Netherlands
[4]Division of Occupational Therapy, Department of Neurobiology, Care Sciences and Society, Karolinska Institutet, Stockholm, Sweden
[5]INSERM, University of Toulouse, Toulouse, France
[6]Department of Epidemiology and Public Health, Toulouse University Hospital, Toulouse, France
[7]Aging Research Center, Department of Neurobiology, Care Sciences and Society, Karolinska Institutet and Stockholm University, Stockholm, Sweden
[8]Section of Medical Ethics, Department of General Practice, Amsterdam UMC/University of Amsterdam, Amsterdam, The Netherlands
[9]Department of Neurology, Amsterdam UMC/University of Amsterdam, Amsterdam, The Netherlands
[10]Department of Neurology, Donders Institute for Brain, Cognition and Behaviour, Radboud University Medical Center, Nijmegen, The Netherlands
[11]Neurocenter Finland, Department of Neurology, Kuopio University Hospital, Kuopio, Finland
[12]Theme Aging, Karolinska University Hospital, Stockholm, Sweden
[13]R&D Unit, Stockholms Sjukhem, Stockholm, Sweden
[14]Neuroepidemiology and Ageing Research Unit, School of Public Health, Imperial College London, London, United Kingdom
[15]Institute of Public Health and Clinical Nutrition, University of Eastern Finland, Kuopio, Finland

**Acknowledgements** The authors thank all the study participants, Carin Miedema and Suzanne Ligthart, for their assistance in the organisation and moderation of the Dutch focus groups; Floor Rooskens and Tessa van Middelaar for their assistance in the coding and analysis of the Dutch transcripts; and Shireen Sindi and Sophie Gaber for the language edition.

**HATICE Study group** The members of the HATICE study group are: Cathrien Beishuizen, Edo Richard, Eric Moll van Charante, Lennard van Wanrooij, Marieke Hoevenaar-Blom, Pim van Gool, Susan Jongstra, Tessa van Middelaar, (Academic Medical Center, University of Amsterdam, Amsterdam, The Netherlands); Hilkka Soininen, Mariagnese Barbera (University of Eastern Finland, Kuopio, Finland); Tiia Ngandu (Finnish Institute for Health and Welfare, Helsinki, Finland); Francesca Mangialasche, Miia Kivipelto (Karolinska Institutet, Stockholm, Sweden); Juliette Guillemont, Nicola Coley, Sandrine Andrieu (INSERM-Toulouse University, Toulouse, France); Yannick Meiller (Novapten, Paris, France); Bram van de Groep (Vital Health Software, Ede, the Netherlands); Carol Brayne (University of Cambridge, Cambridge, UK).

**Contributors** Obtained funding: MK, ER, EMvC, HS and SA; Study design: UA, CRLB, MB, EMvC, FM, ER, HS, SA, MK, AJP, JG, NC and SS. Data collection: CRLB, MB, AR, LS, JG, NC and SS. Data analysis: UA, CRLB, MB, EMvC, JG and SS. Interpretation of the results: UA, MFP, MB, CRLB, EMvC, AR, FM, NC and AJP. Drafting of the manuscript: UA, MFP andMB. The manuscript has been approved by all the authors.

**Funding** The research leading to these results has received funding from the European Union Seventh Framework Programme (FP7/2007-2013) under grant agreement no 305 374. The study has also been funded by the Joint Programme Neurodegenerative Disease (JPND) 'Multimodal preventive trials for Alzheimer's Disease: towards multinational strategies-programme: MIND-AD,' Academy of Finland (291803) and VTR, Kuopio University Hospital (5772815), Swedish Research Council (529-2014-7503), The Stockholms Sjukhem foundation, The Netherlands Organisation for Health Research and Development, (733051041) and The French National Research Agency (ANR-14-JPPS-0001–02).

**Competing interests** None declared.

**Patient and public involvement** Patients and/or the public were not involved in the design, or conduct, or reporting, or dissemination plans of this research.

**Patient consent for publication** Not required.

**Ethics approval** The HATICE study was approved by the medical ethics committee of the Academic Medical Center (Netherlands), the Comité de Protection des Personnes Sud Ouest et Outre Mer (France), and the Northern Savo Hospital District Research Ethics Committee (Finland). For the present substudy, no separate approval was required in Finland and France; in the Netherlands, approval was obtained from the medical ethics committee of the Academic Medical Center.

**Provenance and peer review** Not commissioned; externally peer reviewed.

**Data availability statement** In order to guarantee confidentiality and anonymity of the participants, data cannot be made publicly available. For more information please contact the corresponding author.

**ORCID iDs**
Ulrika Akenine http://orcid.org/0000-0002-7198-9368
Juliette Guillemont http://orcid.org/0000-0002-8608-6751

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
