## [Reviewer comments · BMJ Open]

ARTICLE DETAILS

TITLE (PROVISIONAL)	Attitudes of at-risk older adults about prevention of cardiovascular disease and dementia using eHealth: a qualitative study in a European context
AUTHORS	Akenine, Ulrika; Barbera, Mariagnese; Beishuizen, Cathrien; Fallah Pour, Mandana; Guillemont, Juliette; Rosenberg, Anna; Coley, Nicola; Mangialasche, Francesca; Salo, Lotta; Savy, Stephanie; Pols, A.; Andrieu, Sandrine; Richard, Edo; Soininen, Hilikka; Moll van Charante, Eric; Kivipelto, Miia

VERSION 1 – REVIEW

REVIEWER	Lisanne Kremer Niederrhein University of Applied Sciences, Faculty of Health Care, Krefeld, Germany
REVIEW RETURNED	07-Feb-2020

GENERAL COMMENTS	 - The background should be described in more detail. Research project? Previous Research? - It would be helpful to integrate an objective section to the paper, too. (not only in the abstract) - There are some unclear points due to language difficulties. - Please have a closer look on your methods section. Especially on the description of the focus group as well as on design. It is not quite clear how the Focus group was structured in terms of content.
--

REVIEWER	Wouters, EJM Fontys University of Applied Science, School of allied health professions (1) and Tilburg University, School of social and behavioral science, dpt. TRanzo
REVIEW RETURNED	28-Feb-2020

GENERAL COMMENTS	This is an interesting paper that explores persons with CVD and dementia risk factors to use eHealth. It is valuable that the results are retrieved and 'generalised' from three West-European countries. I have some major and minor remarks to consider though. Most important is the actual research question. Reading the title of the article, it is suggested that the main research question here is on the willingness to participate in preventive eHealth. However, it seems that knowledge and attitudes towards the diseases is also a study topic. In my opinion, they are two different questions to be addressed seperately. I recommend to reconsider your choice in this respect. Another choice that is not yet addressed in the study, is the choice for both CVD and dementia. Especially as one of the major themes is 'burden and stigma of dementia', this choice (to include CVD and
---

	dementia prevention in the same study) needs further explanation. In general: it is not quite clear whether the paper is about primary, secondary (or even tertiary, as there are also (table 3) persons who actually have CVD) prevention. How is prevention defined here and what are the risk factors considered as inclusion criteria? (see: attachment – Please contact publisher for this file)
--	---

VERSION 1 – AUTHOR RESPONSE

REVIEWER 1 EVALUATION:

1) The background should be described in more detail. Research project? Previous research?

Author response: more clarifications has been done in the background by adding previous research.

2) It would be helpful to integrate an objective section to the paper too, (not only in the abstract).

Author response: The aim has been revised and can be found at the end of background (p. 6-7).

3) There are some unclear points due to language difficulties.

Author response: The language has been checked and edited by a professional translator.

4) Please have a closer look on your methods section. Especially on the description of the focus group as well as on design. It is not quite clear how the focus group was structured in terms of content.

Author response: More clarifications have been done in the data collection regarding the content of focus groups as recommended (p. 9-10) and also design, recruitment and inclusions criteria (p. 7-8).

REVIEWER 2 EVALUATION:

General remarks

1) Most important is the actual research question. Reading the title of the article, it is suggested that the main research question here is on the willingness to participate in preventive eHealth. However, it seems that knowledge and attitudes towards the diseases is also a study topic. In my opinion, they are two different questions to be addressed separately. I recommend to reconsider your choice in this respect.

Author response: The aim of the study has been revised as recommended focusing (p. 6-7).

2) Another choice that is not yet addressed in the study, is the choice for both CVD and dementia. Especially as one of the major themes is 'burden and stigma of dementia', this choice (to include CVD and dementia prevention in the same study) needs further explanation.

Author response: This point has been further clarified in the background as recommended (p. 5).

3) In general: it is not quite clear whether the paper is about primary, secondary (or even tertiary, as there are also (table 3) persons who actually have CVD) prevention. How is prevention defined here and what are the risk factors considered as inclusion criteria?

Author response: This point has been further clarified in the background (p. 5) and participants and descriptions for the inclusion criteria as recommended (p. 7-8).

Remarks per section

Introduction

1) P. 3 line 54: second 'it' in sentence refers to 'positive attitude'? Please make this explicit.

Author response: This point has been clarified and revised in the text (p. 5).

2) P. 4, please add why it is important in your opinion to consider CVD and dementia together.

Author response: This point has been clarified and revised in the text (p. 5).

3) Research question: the knowledge and attitude towards prevention is, in itself two research questions as we know that knowledge and attitudes are not two sides of the same coin. Moreover, engaging in eHealth programmes, has another specific dimension: namely, the acceptance and capability (among many others) to use digital technology. Why are they combined in one study (please explain in the introduction and provide the reader with some theoretical background on your considerations)?

Author response: The aim has been revised as suggested focusing on attitudes in line with the results (pages 6-7). The title, abstract and background has been revised accordingly. Background has been developed referring to the related literature (p. 6).

Methods

1) P. 4, line 41-46: this sentence is not very clear: why would '....expertise in qualitative, clinical and basic science research' ensure '(an international perspective and) a thorough understanding of the local healthcare settings. Please rephrase the sentence.

Author response: The sentence has been rephrased and revised (p. 7).

2) Please provide a bit more detail on the inclusion criteria and the choices made in this respect.

Author response: More information is provided to describe the inclusion criteria as recommended (p. 7-8).

3) I appreciate the thoroughness of the focus groups (number et cetera). Could you also explain what this procedure (two extra focus groups in Finland and The Netherlands) was based upon (why did you do it this way)? Furthermore: what is meant by 'technical difficulties' and analysis of detailed notes' and by '...allowed to freely develop the discussion'?

Author response: Clarifications has been done in descriptions of the focus groups in data collection (p. 9) and data analysis (p. 10), and even discussed in the methodological considerations (p. 21).

4) From p. 6 onward, you write about a topic list. What exactly do you mean? Normally (i.e., in semi-structured interviews) a topic list would be constructed from literature and used during the interviews. Then I would expect the theory base in the introduction. You on the other hand performed open interviews (using grounded theory): how exactly did you work and how does a topic list comes in here?

Author response: Clarifications has been done in descriptions of the focus groups in data collection (p. 9) and Table 2.

5) P. 6, l. 19: what about member checking in the other focus groups?

Author response: Clarifications has been done in descriptions of data analysis as recommended (p. 10) and further discussed in the methodological considerations (p. 21).

6) P. 6, l. 31: (see earlier remark): how were patients included exactly: i.e., how was 'at risk for CVD or dementia' established?

Author response: More information is provided to describe the inclusion criteria as recommended (p. 7-8).

Results

7) P. 6, line 39: here it seems that the research question was solely directed towards attitudes of persons at risk for CVD an dementia regarding prevention using eHealth. So this should be more clear from the start (the introduction).

Author response: The title, abstract, introduction, and aim, have been revised according to reviewer's comments and in line with the results with a focus on attitudes of at-risk older adults about prevention of CVD and dementia.

Discussion

8) P. 12, line 41-42: here the aim is (again) slightly differently described: 'the aim was to explore the knowledge and attitudes towards prevention among older adults at increased risk of CVD and dementia'. Please make the aim of the study more consistent in all parts of the paper.

Author response: The aim has been consistently revised in the whole manuscript (abstract, introduction and the beginning of discussion) as recommended.

9) P. 12, l. 48 'Additionally, a core category was developed: the interactive process of the three identified categories influencing engagement in self-management prevention programmes. The three categories were interconnected through an interactive process...'. Here it seems two times the same is stated, but not exactly explained: how are these processes interconnected? Is there (perhaps in other domains) already some literature available to explain this?

Author response: The core category is the process of how the three categories are interacting with each other. The description of this process is described and presented in results section on pages 15-16; however, this process has been further clarified in the discussion as recommended (p. 16-17). Moreover, the related literature has been added to this section as recommended by the reviewer (p.

17).

10) P. 12, line 55: 'influenced by the local health care culture'. The core theme (and especially this part of it) is not illustrated by quotes in the results section: for validity, this should be added. For the discussion: can you explain (discuss) this a bit more? Also, no information is given on differences of the system. Only at the end of the discussion, some information is provided, which seems not logical.

Author response: The core category was developed by the three categories. The core category is about how the three categories interact with each other; Two quotes are presented in the related section in the results where the core category has been presented (p. 15-16). The healthcare culture and context is the background or the setting in this study. The aim of the study is not how these three healthcare systems are different from each other, however as suggested by the reviewer the differences in primary healthcare systems have been described in the methods as setting (p. 8-9) and further described in connection to the related literature in the discussion (p. 20). The interactive process of the three categories, describing the core category has been revised in the discussion (p. 16-17) and was connected to the literature (p. 17).

11) P.13, l. 10: '....and have a relationship of trust with their healthcare provider': should this be 'or' or perhaps and/or (instead of and, as this doesn't seem to apply to the Finnish participants so much as to the other nationalities).

Author response: This point has been clarified.

12) P. 13, l. 42: '...the health care system': see earlier remarks. This conclusion should be illustrated and explained more in both the results and the discussion section. It would be better to describe these differences as well (in the introduction or methods section). Now it is only casually mentioned in the end of the Discussion section!

Author response: The description of the setting has been added in order to describe shortly the differences of healthcare systems in three countries (p. 8-9). It has also been discussed in the discussion as suggested (p. 20).

13) P. 13, l. 56: 'the lack of knowledge about dementia': lack of knowledge by whom?

Author response: This point has been clarified in the discussion (p. 18).

Minor remarks

- l. 16 (abstract): within research group; should this be: 'within THE research group or', alternatively 'within research groups'?

Author response: This point has been clarified in the abstract (p. 3).

- l. 9 (Introduction): delete 'cases'.

Author response: This point has been clarified in the introduction (p. 4).

- There is sometimes a space between word/ punctuation mark and reference (number), sometimes it lacks (e.g., 15 versus 23)

Author response: This point has been clarified in the manuscript; the manuscript has been checked by a language editor.

- Method, p. 5, l. 50: dot is missing after 'consensus'.

Author response: This point has been revised in the methods (p. 9).

- Results, p. 11, line 18: 'trust in' delete 'in'

Author response: The language has been checked by a native British- English editor; the study by the approve of the language editor has used the term "trust in" to present what the participants experienced.

Tables

- Ad table 3: shouldn't the title include the worth 'eHealth'?

Author response: Table 3 has been removed from the manuscript since the categories and the core category have been presented in the figure 2.

- Ad table 1: also persons with a history of CVD were included. What kind of history (very broad now) and: this was not clear in the Methods part. This also stresses the importance to define 'prevention of CVD and dementia': what kind of risk factors, how have participants been selected?

Author response: The information about inclusion criteria, including the person with a history of CVD has been developed in the methods, participants (p. 7-8) which further clarifies the reviewer's point.

- Note: you give means here: is there a normal distribution of the variables to justify giving means (instead of medians)?

Author response: The mean amount has been presented in Table 1. No analysis has been performed to present the normal distribution since this study is a qualitative study.

Figures

- Figure of the themes could be skipped: does not provide much extra information.

Author response: Table 3 has been removed and Figure 2 has been further developed and presented in the manuscript.

VERSION 2 – REVIEW

REVIEWER	Lisanne Kremer Niederrhein University of Applied Sciences
REVIEW RETURNED	03-Apr-2020
GENERAL COMMENTS	thank you for your detailed revision of the manuscript. The common thread is now clearly recognizable and methodological difficulties that I noted are clarified in a manner that makes them comprehensible and plausible.
REVIEWER	Eveline Wouters Fontys UAS and Tilburg University
REVIEW RETURNED	14-Apr-2020

GENERAL COMMENTS

The authors have addresses the feedback given in a satisfactory manner, and have improved their paper sufficiently.